# Gain of Function (GOF) Mutant p53 in Cancer—Current Therapeutic Approaches

**DOI:** 10.3390/ijms232113287

**Published:** 2022-10-31

**Authors:** Katarzyna A. Roszkowska, Aleksandra Piecuch, Maria Sady, Zdzisław Gajewski, Sylwia Flis

**Affiliations:** Center for Translational Medicine, Warsaw University of Life Sciences, 100 Nowoursynowska St., 02-797 Warsaw, Poland

**Keywords:** mutant p53, gain-of-function, targeted therapy, p53 reactivation, review

## Abstract

Continuous development of personalized treatments is undoubtedly beneficial for oncogenic patients’ comfort and survival rate. Mutant *TP53* is associated with a worse prognosis due to the occurrence of metastases, increased chemoresistance, and tumor growth. Currently, numerous compounds capable of p53 reactivation or the destabilization of mutant p53 are being investigated. Several of them, APR-246, COTI-2, SAHA, and PEITC, were approved for clinical trials. This review focuses on these novel therapeutic opportunities, their mechanisms of action, and their significance for potential medical application.

## 1. Introduction

The tumor suppressor p53, encoded by the *TP53* gene and known as “the guardian of the genome” [1], performs a variety of functions in cancer prevention. The basic unit of the p53 protein consists of three major functional domains such as an N-terminal transactivation domain (TAD), a core DNA-binding domain (DBD)—the main target for mutations, and a C-terminal regulatory domain (CTD) (Figure 1A) [2]. The p53 protein, to exert its function, binds in a sequence-specific manner to the DNA-binding sites by forming a tetramer, via four self-assembling p53 molecules, which are stabilized by protein–protein and base-stacking interactions [3]. As a gene-expression regulator, it mainly controls how the cell behaves under stress conditions. p53-specific responses consist of the activation of mechanisms such as cell cycle arrest, apoptosis, and senescence [4,5,6,7]. Responsible for such pivotal processes, p53 can deliver considerable damage if mutated, thus becoming “the guardian of the cancer cell” [8]. *TP53* is the most frequently mutated gene in cancers. It has been recognized that 50% of cancer patients acquire certain types of *TP53* gene alterations [9,10]. The potential role of p53 as a specific target in modern therapies against cancers is being widely discussed. Numerous attempts at this approach have already been made. *TP53* mutations are labeled as loss of function (LOF) or gain of function (GOF). The presence of GOF *TP53* mutants increases the malignancy of tumors in various ways. Occurring metastases, greater chemoresistance, invasiveness, and shorter survival are typical traits of GOF. These mutations account for 30% of all missense mutations in the *TP53* gene. Widespread screening of patients allowed for the recognition of the hotspots R175, G245, R248, R249, R273, and R282, which are present in the DNA-binding domain (DBD) or near this interface of p53 [11].

## 2. Pivotal Functions and Regulation Mechanisms of Wild-Type p53 (WTp53)

Non-altered p53 acts as a transcription regulator in charge of the cellular responses to stress factors, hypoxia, nutritional stress, differentiation signals, and DNA damage. Under stress conditions, an affected cell can respond in numerous ways. p53 induces cell cycle arrest, apoptosis, or senescence. Each of those processes is regulated by p53 and its target genes. Cell cycle arrest is induced by p21 and p57; apoptosis is activated via Puma, also known as BCL2 binding component 3, Bax (BCL2-associated X), and Noxa (phorbol-12-myristate-13-acetate-induced protein 1); and senescence is induced via p27^Kip1^ (cyclin-dependent kinase inhibitor 1B, encoded by *CDKN1B* gene) and Pai1 (phosphoribosylanthranilate isomerase 1), while TIGAR (*TP53* induced glycolysis regulatory phosphatase) and glutaminase 2 (encoded by *GLS2*) are responsible for the arrangement of metabolic changes [7,15,16]. Additionally, p53 upregulates the expression of death receptor DR5 and, thus, may mediate apoptosis in part via DR5 [17]. The negative regulation of p53 activity is dependent on the Mdm2 protein, which binds to p53 and, as a result, inhibits its transcriptional functions, thus promoting proteasomal degradation. WTp53 oncogenic suppressor functions are regulated by the presence of molecular chaperones, which shape the proper tertiary and quaternary structure of the protein [18,19] However, mutant p53 (MUTp53) benefits from this mechanism, due to the chaperones that stabilize MUTp53, thus allowing it to escape proteasomal degradation [20]. It is a well-known fact that MUTp53 is abundant in cancer cells compared to WTp53 in non-tumorigenic cells, which indicates that the mutated protein is more stable [21]. Although MUTp53 undergoes the degradation mechanisms orchestrated by Mdm2 in normal tissues, the same process fails in tumors for an unknown reason [22]. Several studies were conducted to explain this phenomenon. Cancer tissues have the tendency to express *HSP70* or *HSP90*, which stabilize MUTp53, therefore allowing its accumulation in the cell [23]. However, this explanation undermines the regulatory patterns, including those controlled by Mdm2 or other E3 ligase proteins [24].

## 3. Features of GOF p53 Mutants

Missense mutations occurring in the six hot spots consist of eight mutants, which account for nearly 30% of all missense mutations. They consist of R175H, G245S, R248Q, R248W, R249S, R273C, R273H, and R282W. The gain of function characteristics are designated to all of them, although the mechanisms behind their novel functionalities are different. This group can be further divided, based on the dysfunctionality that is present in the final protein. Mutation in the contact mutants, R248Q, R248W, R273H, and R273C, occurs in their DNA-binding domain, directly affecting their ability to control the transcription of targeted genes. Conformational mutants such as R175H, G245S, R249S, and R282H, on the other hand, are unable to fold properly, leading to the loss of the zinc coordinates and, thus, general DNA-binding activity [25,26,27,28,29,30]. Contact mutants and conformational (structure) mutants show decreased thermostability. Altered proteins are not capable of binding to the designated sites; however, they are capable of binding to new sequences, can regulate completely different genes, as a result, and produce new phenotypes [30]. Consequently, each of the substitutions can affect the newly formed protein differently; hence, each mutant promotes distinct GOF hallmarks caused by individual molecular mechanisms, which may require a novel approach in therapy. The clinical approach is highly complicated, because many factors must be taken into account; nevertheless, the result could notably benefit the condition of cancer patients [7,31,32]. More than 40% of cancer-related gene expression depends on the SWI/SNF signaling pathway. SWI/SNF is a subfamily of ATP-dependent chromatin remodeling complexes, which serve broad roles in the transcriptional regulation of differentiation and proliferation across many lineages. Lately, it has been shown that MUTp53 can interact with the SWI/SNF complex, resulting in chromatin being in an open state due to the histone modifications. As a consequence, changes in the expression of cancer-related genes were observed [33]. Other proteins undergo changes in their activity via MUTp53, such as p63 and p73 [34], which, in turn, inhibit apoptosis instead of inducing it. Various other transcription factors are under the influence of MUTp53 both in negative and positive regulatory activity: ETS2, NF-kB (nuclear factor kappa B), HIF-1α (hypoxia-inducible factor 1 subunit alpha), SMAD, SREBP (sterol regulatory element binding protein), and NF-Y (nuclear transcription factor Y) [29].

## 4. Chemoresistance Mechanisms Established by MUTp53

Cancer cells’ ability to manage oxidative stress is possible due to the presence of the xCT (also known as SLC7A11), a functional subunit of the cystine/glutamate antiporter system xc-. xCT is coded by the *SLC7A11* (solute carrier family 7, member 11) gene, and its overexpression is observed in various types of cancer cells, especially in cancer stem cells (CSC). xCT alters metabolic pathways via participation in glutathione biosynthesis and, in this way, protects cancer cells from oxidative stress conditions and ferroptosis [35]. Generally, the presence of xCT in cells is associated with the promotion of tumor progression and induction of chemoresistance through the detoxification activity of GSH-mediated reactive oxygen species (ROS). It was shown that WTp53- as well as p53-carrying GOF missense mutations can inhibit the expression of *SLC7A11* and sensitize cells to ferroptosis. It was also shown that cells with non-functional p53 are highly resistant to chemotherapeutics and radiotherapy; however, the knockout of the *SLC7A11* gene results in the restoration of sensitivity to applied therapy. These observations are related to GSH depletion and, consequently, to the reduced protection from oxidative stress upon xCT inhibition. This suggests that, in the resistance to oxidative stress, the regulation of the xCT-glutathione axis plays an important role, which allows the tumor to survive in unsuitable conditions. Therefore, the inhibition of the xCT-glutathione axis may represent a promising approach to overcoming resistance associated with MUTp53 [36,37]. This significantly increases the resistance to therapeutic drugs, such as cisplatin, adriamycin, and etoposide, even when compared to cancer cell lines with *TP53* knockdown [38]. Etoposide resistance was also observed in the study conducted by Scian et al., where it was linked to the increased expression of NF-κB2 induced by MUTp53^R273H^ and ^R175H^. Interestingly, the mutant D281G did not cause such effects [39]. Cisplatin resistance can be overcome in the mutant R273H via depletion of ataxia-telangiectasia (ATM) and the Rad3-related protein (ATR) activator DNA2 [40]. The understanding of these mechanisms is crucial for the successful treatment of patients. Overall, studies show that MUTp53 is involved in the increased expression of *MDR1* (multidrug resistance gene 1) [41]. Each type of GOF MUTp53 can manifest several chemoresistance mechanisms due to different proteins and genes being affected. For instance, R273H is resistant to doxorubicin and methotrexate via the inhibition of apoptosis through procaspase-3 downregulation [42]. The resistance to gemcitabine occurs due to MUTp53 phosphorylation, which induces *CDK1* (cyclin-dependent kinase 1) and *CCNB1* (cyclin B1) expression [43], while R273H mutant is resistant to cisplatin via YAP/β-arrestin1 pathway [44]. Long non-coding RNAs (lncRNAs) are associated with chemoresistance and the proliferation of tumors. Cells carrying the R273H mutation were established to have more in common with CSC than other mutants. Moreover, lnc273-31 and lnc273-34 were required for CSC to establish the self-renewal feature. Generally, epithelial–mesenchymal transition (EMT), migration, invasion, and chemoresistance were established as the characteristics of R273H mutant cells, which demonstrate the high expression of lnc273-31 and lnc273-34. However, this effect was not manifested in R175H or R248W p53 mutants [45]. Another noteworthy chemoresistance mechanism involves the Mdm2-mediated ubiquitination and degradation of the mutant p53. It was observed that the p53^R248Q^ mutant’s resistance to cisplatin could be modulated by fibroblast growth factor-inducible 14 (Fn14). High-grade serous ovarian cancer cells became sensitive to cisplatin, due to p53^R248Q^ degradation, which was possible when expression of Fn14 was restored [46].

## 5. Possible Therapeutic Approaches

Novel strategies in the therapies targeting MUTp53 have been presented over the past decade. These strategies include the elimination of mutant p53 and restoration or reactivation of WTp53, destabilization of MUTp53, or inhibition of the downstream signaling resulting from mutant p53 gain of function and, thus, initiation of synthetic lethality in the cells expressing mutant p53. Numerous compounds targeting MUTp53 have already been discovered, and various brilliant reviews have already described them in a comprehensive manner [26,47,48,49,50,51]. However, only a few of these compounds have reached the clinical stage of research. This review focuses on the current therapeutic opportunities for oncologic patients. The molecules taken into consideration for this article are APR-246 (PRIMA-1MET, eprenetapopt), COTI-2, vorinostat (SAHA), and PEITC (phenethyl isothiocyanate). All of the reviewed molecules are summed up in Table 1.

## 6. APR-246 (PRIMA-1MET, Eprenetapopt)

APR-246 was discovered by cell-based screening in 2002 by Bykov and collaborators [52]. This molecule is a prodrug that, after proper conversion, targets the conformational p53 mutants and restores their native form [56], therefore restoring their transcriptive functions and allowing them to regulate the expression of targeted genes, such as *PUMA*, *NOXA,* and *BAX,* which, in consequence, promotes apoptosis [52,57].

APR-246 is transformed into a reactive compound methylene quinuclidinone (MQ), which is capable of covalent binding to the thiol groups in MUTp53 and WTp53 [58]. This process restores the MUTp53 DNA-binding ability. Depending on the residue, where the MQ molecule binds, it displays different mechanisms. For instance, the MQ binding to the 277-cysteine residue stabilizes the p53-DNA interface. MQ-C124 (cysteine residue 124) and MQ-C229 (cysteine residue 229) support the interface between p53 dimers. Overall, these processes function as support for the p53-DNA complexes [59]. APR-246 cytotoxicity is induced due to the accumulation of ROS. MUTp53 itself suppresses the expression of *SLC7A11* by targeting *NRF2* (nuclear factor erythroid 2-related factor 2), which prevents the formation of the antioxidant-glutathione (GSH). MQ, additionally, leads to ROS abundance by binding to the thiol groups of GSH [57,60,61]. This topic was further investigated by Milne and co-workers using a human cell line of non-small cell lung cancer H1299 with mutant p53^R175H^ or ^R273H^. The authors revealed that this mechanism disrupts the functioning of the R175H mutant but not of R273H. APR-246 demonstrates higher toxicity, when *SLC7A11* is downregulated, but only for the R175H mutant and not for R273H [62]. This indicates that some mutants are more vulnerable to APR-246, while others are not. Behind these processes, a different mechanism is in charge, which results in distinct sensitivity to the applied drug. However, some studies claim that the effectiveness of APR-246 is not dependent on the p53 status [63,64,65]. Despite former research, *TP53* mutation status may not be the best predictive factor for APR-246 sensitivity. Recently it was concluded that the *SLC7A11* expression is a significantly more precise factor. Additionally, the *SLC7A11* genetic regulators, such as ATF4, Mdm2, WTp53, and c-Myc, modulate the cancer resistance to APR-246 [65]. Intriguingly, some results show that cell line CCRF-SB with WTp53 is particularly resistant to APR-246 [66]. Nevertheless, APR-246 is a promising therapeutic in combination with other clinically used chemotherapeutics. APR-246 affects cells with *TP53* mutations, such as OVCAR-3^R248Q^ (human ovarian cancer), NSCLC^R248W^, and ^R273H^ (non-small cell lung carcinoma), to become sensitive to doxorubicin and cisplatin [57]. Different types of mutation significantly impact the response to the applied therapy. In the pancreatic ductal adenocarcinoma cell line with GOF *TP53* (R248W cell line), it was discovered that introducing WTp53 increased the sensitivity to APR-246 [67].

APR-246 is currently registered for 13 clinical trials in patients suffering from various cancers (Table 2). Within those clinical trials, some results are promising. The use of eprenetapopt in patients diagnosed with hematologic malignancies and solid tumors has been proven safe [68,69,70,71], while, in another study, it was confirmed to give better results when combined with azacitidine [72], even greater than azacitidine alone [73], in patients with leukemias.

## 7. COTI-2

COTI-2 is a third generation of thiosemicarbazone that targets p53 mutants and restores their native conformation. The molecule was registered for clinical trials and further studied in 2016 by Salim and co-workers via machine learning in silico screening (computational platform known as CHEMSA) [53]. COTI-2 is a Zn2+ chelator that binds to the misfolded mutant p53 and restores its proper folding [53,74]. COTI-2 is effective independently of p53, and its mechanism involves the activation of AMPK and inhibition of the oncogenic mTOR pathways, which suggests the presence of other targets. Its actions result in cell senescence rather than apoptosis, probably due to its effect on p21 [75]. It was reported that COTI-2 is safe in animal models [53].

The efficacy of this therapeutic agent has been tested on multiple cell lines and mice xenografts. Mostly, it demonstrated antiproliferative activity, greater than the one observed with the use of cetuximab or erlotinib, both of which were already approved for cancer treatments. COTI-2 was tested on human cell lines with different kinds of *TP53* mutations, HT-29 (colon cancer, p53^R273H^), HCT-15 (colorectal adenocarcinoma, p53^S241F^), OVCAR-3 (ovarian carcinoma, p53^R248Q^), K562 (chronic myelogenous leukemia, p53^Q136fs*13^), SF-268 (glioblastoma, p53^R273H^), SNB-19 (glioblastoma, p53^R273H^), T47D (breast cancer, p53^L194F^), MDA-MB-231 (breast cancer, p53^R280K^), *KRAS* mutations (MDA-MB-231, colorectal cancer cell line SW620 with p53^R273H^), *PIK3CA* mutations (breast cancer cell line MCF7 with WTp53, HT-29, T47D), *APC* mutations (colon cancer cell line COLO-205 with p53^Y103F^, HCT-15), and *PTEN* mutations (glioblastoma cell line SF-295 with p53^R248Q^, SNB-19), which showed its therapeutic potential by inhibiting the growth of all the above mentioned cell lines. The results of the study conducted by Salim and co-workers offered several promising discoveries. COTI-2 was also compared to approved therapeutics such as cetuximab, erlotinib, cisplatin, and carmustine on various colon cancer and glioblastoma cell lines, in which it was proven to be more effective [53]. COTI-2 was also evaluated as an additional therapeutic component. Lindemann and co-workers demonstrated how additional usage of COTI-2 resulted in a higher sensitivity to cisplatin and radiation in head and neck squamous cell carcinoma (HNSCC), regardless of *TP53* status [76]. Another study showed that cells with MUTp53 were more sensitive to COTI-2 than cells with WTp53. However, COTI-2’s effectiveness was present in all types of p53 mutations, regardless of whether they were a conformational or a contact mutant. Additionally, in this study, COTI-2 was also effective independently of p53. Cells, upon receiving the treatment, demonstrated greater p63 levels and p63’s enhanced binding to the promoters of p21 and Puma [77].

There is one registered clinical trial for COTI-2. This is a phase I study of COTI-2, as a monotherapy or in combination with cisplatin (Table 3). The patients with HNSCC must have confirmed *TP53* mutations; however, the recruitment status is currently unknown.

## 8. Vorinostat (SAHA)

Suberoylanilide hydroxamic acid (SAHA, Vorinostat) is a histone deacetylase inhibitor (HDACi). Originally it was used for cutaneous T-cell lymphoma treatment. SAHA was already approved by the FDA; however, more possibilities for its use in cancer therapy are continuously under development. Recently, it was shown to target MUTp53 specifically and induce its degradation [54]. Primarily, it regulates the acetylation of proteins including nucleosomal histones. SAHA induces apoptosis, as a result of cytochrome c release and ROS accumulation via the mitochondria-mediated pathway. In consequence, SAHA does not require functional p53 [78]. Nevertheless, p53-mediated apoptosis after treatment with SAHA has also been reported in another study [78,79].

SAHA induces cell senescence, independently of MUTp53 status. The mechanism is still under examination. Some studies suggest that it promotes MUTp53 degradation selectively, in this case in MDA-MB-231^R280K^ and DLD1^S241F^ (colorectal adenocarcinoma); however, SAHA-induced cell death was only present in the MDA-MB-231 cells and not in DLD1 [80].

Additionally, SAHA is capable of inducing apoptosis in cells that lack p53 mutations in a mechanism completely independent from p53. The mechanism consists of p21^WAF1/CIP1^ elevation via the inhibition of Mdm2, but only in the LNCaP (human prostate adenocarcinoma) cell line and not in MCF-7, despite elevated levels of p53 and p27^Kip1^ [81,82]. A different study conducted by Drozdkova and co-workers also suggested the p53-independent mechanism of SAHA in cancers. This is due to the apoptosis occurring in the tested cell lines with a mutation in the *TP53* gene (U266^A161T^ and RPMI8226^E285K)^, although it had a greater impact on RPMI226 cells [83].

The study conducted by Huang and co-workers suggests a direct reaction between SAHA and p53, in cases where apoptosis was a result of p53 activation via phosphorylation. This research was carried out on an in vitro model, where nasopharyngeal carcinoma cells showed that SAHA activates tumor suppressors such as p53 and Rb1 (retinoblastoma protein), while, at the same time, inactivating AMPK (5′ AMP-activated protein kinase) signaling, which leads to apoptosis [84]. SAHA is highly effective in cell lines with p53 mutations, breast cancer, MDA-MB-231^R280K^, BT-474^E285K^, and prostate adenocarcinoma PC3^p.K139fs*31^, where the antiproliferative effect was present due to the increased expression of *CDKN1A* (cyclin-dependent kinase inhibitor 1A encoding p21), while, at the same time, *CCND1* (cyclin D1) and *TP53* expression levels decreased [85].

Successful in a variety of cancers in in vitro research (lymphoma, myeloma, leukemia, mesothelioma, colon carcinoma, NSSCLC, bladder, breast, prostate, ovarian, renal cell, thyroid, pancreatic, endometrial cancer, melanoma, glioblastoma) and well-tolerated by patients diagnosed with cutaneous T-cell lymphoma, SAHA has established its multipurpose use [86]. In the clinicaltrials.gov database, there appears to be 354 registered studies when using the phrase “SAHA” or “vorinostat” in the research table. The majority of those studies investigate the opportunities of cancer therapy; however, other conditions such as HIV infections (NCT01365065, NCT02707900, NCT03803605), Cushing’s disease (NCT04339751), sickle cell diseases, anemias (NCT01000155), Niemann–Pick disease (NCT02124083), and Alzheimer’s disease (NCT03056495) can also be found.

Selected clinical trials of SAHA, whether used alone or in combination with other drugs, cover a variety of cancers, and the range is summarized in Table 4. The effectiveness of vorinostat treatment in neoplasms should be carefully examined due to the quantity of conducted clinical trials.

## 9. PEITC

β-phenylethyl isothiocyanate (PEITC) is a phytochemical that can be found in cruciferous vegetables. This compound acts as a Zn2+ chelator, which inhibits the misfolding of MUTp53. Its therapeutic properties targeting cancer diseases were proposed by Aggarwal et al., who found this compound through cell-based screening [55]. Various studies revealed that PEITC caused oxidative stress, which inhibited the growth of cancer cells [94,95,96,97].

Studies conducted in the mouse breast cancer xenograft model (the SK-BR-3 xenograft mouse) have shown that PEITC selectively targets p53 mutants. This compound acted preferentially towards the p53^R175H^ hotspot mutant and successfully restored its native conformation. In consequence, this mutant was exposed to proteasome-mediated degradation [55]. Moreover, it was shown that PEITC efficiently inhibited tumor growth [55]. In vitro studies have shown the effectiveness of PEITC in treating oral cancer cells with MUTp53 [95]. PEITC is active towards structural (R175H) and contact (R248W) mutants; however, it targets structural mutants favorably, as proven in xenograft prostate cancer mouse models, which resulted in tumor growth inhibition [98]. PEITC affects not only p53 but also other cell-cycle-associated proteins such as CDC25C (M-phase inducer phosphatase 3) and cyclin A2, which cause cell cycle arrest. It was shown that PEITC induces apoptosis in IPEC-J2 cells (intestinal porcine epithelial cell line) by lowering the mitochondrial membrane potential by releasing cytochrome c to the cytoplasm. This cascade of events involves the activation of caspase-9, caspase-3, and PARP 1 (Poly [ADP-ribose] polymerase 1) [97]; therefore, it is still effective in cells lacking the p53 activity. As demonstrated in a study conducted by Liu et al., PEITC lengthened the survival of mice with leukemia. In this experimental model, apoptosis was induced via the decrease in the Mcl-1 (Induced Myeloid Leukemia Cell Differentiation protein) survival molecule, as a result of glutathione depletion and ROS accumulation [99]. Generally, the oxidative stress induced by PEITC and, thus, its effectiveness in eliminating cancer cells, was reported in various studies [96]. The mechanism behind cell cycle arrest in the G2/M phase caused by PEITC application involves oxidative stress in the DNA-damage-induced ATM–Chk2-p53-related pathway [94]. PEITC induces G2/M cell cycle arrest and inhibits the growth of oral cancer cells via the decrease in the Mcl-1 survival molecule, as a result of glutathione depletion and ROS accumulation [99]. Currently, 10 studies with the use of PEITC are registered, and 7 of them include cancer treatment. All of the studies are summarized in Table 5. However, without the published data, it is difficult to determine the legitimacy of the treatment involving PEITC. Nevertheless, one study claims this molecule is potentially effective in inhibiting cancerogenesis in smokers [100].

## 10. Conclusions

Targeted, personalized therapy still has a long way to go, but, with each study, that goal becomes closer. A key question to ask at this stage, of any cutting-edge therapeutic work, is what benefit it may bring to the patient.

The results of the in vitro and in vivo studies of new potential drugs often show the benefits of their use. Unfortunately, at the stage of clinical trials, the expected benefits of treatment often turn out to be far from ideal, especially in oncological patients. When cancer occurs, the changes in p53 are most notably related to its antitumor functions. Not only do mutations of p53 appear but also other abnormalities; thus, specific approaches for such a broad group of patients might be almost impossible. Taking into account variables such as interactions with other genes and metabolites and the dominant negative effects, regardless of whether the mutation is germinal or somatic, the application of personalized therapy in the clinic seems almost impossible [32]. Nevertheless, progress in personalized treatment is still being achieved; for instance, a recent study conducted by Klimovich et al. states that even partial p53 reactivation can induce cancer regression in mice, when the MUTp53^E177R^ variant is considered. However, other mutants have not been studied yet. Researchers point out that when even a partial loss in activity of p53 is introduced, the cancer risk is increased. However, when there is minimal to no activity at all, the phenomena known as p53 addiction [38,101,102] occurs, which means that the cancer cell becomes addicted to the presence of MUTp53 or to the loss of WTp53 activity. The potential medical treatment involves triggering just a small activity of WTp53 through the applied drug, which, in turn, is enough to reduce the tumor [103]. The use of combination treatment when one drug reactivates MUTp53 and the other targets a specific cancer is very promising. As observed in the clinical trial NCT03072043, treatment with eprenetapopt and azacytidine was beneficial to patients with myelodysplastic syndromes and with oligoblastic acute myeloid leukemia, which carries p53 mutants [73]. Other therapeutic approaches such as, for instance, indirect therapy via Mdm2 [31] should also be taken into consideration. Many therapeutics, which are inhibitors of Mdm2 or Mdm4, have entered clinical trials but have yet to be proven to be safe and effective in cancer treatment, such as milademetan, RG7112, RG7388, CGM097, HDM201, and ALRN-6924. Indirect therapy against MUTp53 may prove to be more beneficial and easier to introduce to the clinic, regardless of the specific type of *TP53* mutation. Some recent reviews argue that targeting MUTp53 will provide effective treatment in the future. The development of machine learning technology may come with an easier answer to the future of cancer treatment. Recently a study using such technology reported new genes (*GPSM2, OR4N2, CTSL2, SPERT, RPE65*) that may be associated with p53 functions, which seem to be a better fit for the platinum-based therapies for patients than their *TP53* status [104]. The ongoing discussion between researchers on whether personal therapy, which considers the investigation of molecules targeting the exact type of mutation of *TP53*, should be pursued or not has yet to be unraveled [105,106,107,108,109].

## Figures and Tables

**Figure 1 ijms-23-13287-f001:**
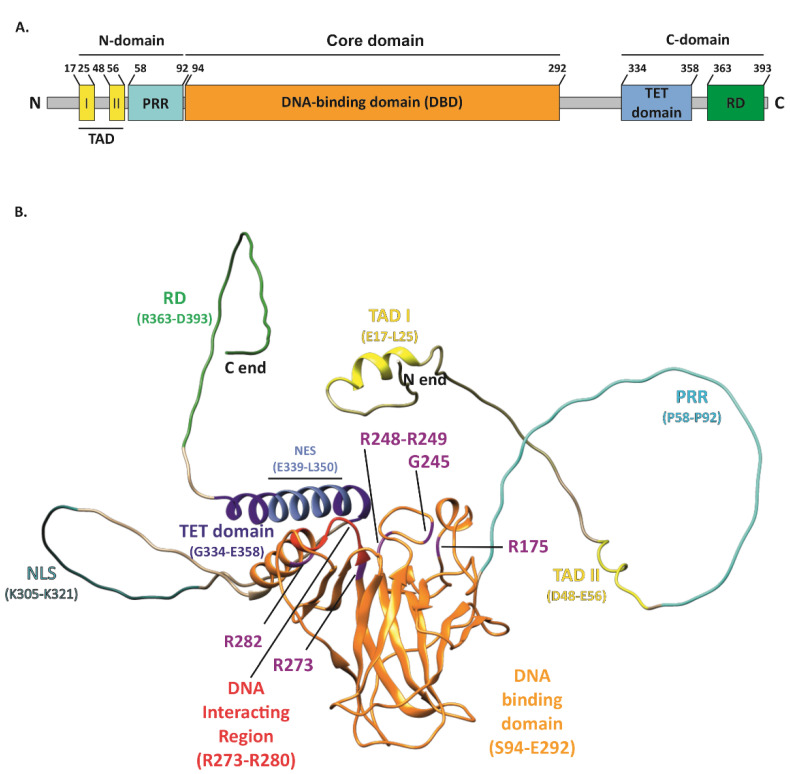
Structure of the p53 protein. (**A**) Simplified representation of the secondary structure showing domain organization of the human p53 protein (Uniprot #P04637). TAD, transactivation domain; PRR, proline-rich region; DBD, DNA-binding domain; TET, tetramerization domain; RD, regulatory domain. (**B**) Schematic representation of the p53 3D structure with the GOF mutation sites shown (purple). The TAD domain is shown in yellow with TAD I and II motifs indicated (yellow); PRR (cyan); DBD (orange) with direct DNA-binding region indicated (red); bipartite nuclear localization signal (NLS) (sea green); TET domain (dark blue) including nuclear export signal (NES) (light blue); and C-terminal RD (green). The structure of p53 was only partially solved by crystallography/X-ray diffraction or NMR; therefore, the 3D structure prediction of full-length protein was performed with AlphaFold DB, DeepMind Technologies Limited [12,13], and visualized with UCSF Chimera, an extensible molecular modeling system developed by the Resource for Biocomputing, Visualization, and Informatics at the University of California, San Francisco [14].

**Table 1 ijms-23-13287-t001:** Therapeutic compounds currently being investigated in clinical trials related to the treatment of cancer.

Drug	Type of Drug	No. of Registered Clinical Trials Related to Cancer Treatment	Mechanism	Targeted p53 Mutants	Phases of Clinical Trials	Discovery
**APR-246** **(Eprenetapopt)**	Small molecule-cysteine thiol group targeting compound	13	Restoration of the native conformation by binding to thiol groups in the core domain	R175H, R273H	I–III	2002 [52]
**COTI-2**	Zn2+ chelator	1	Inhibition of MUTp53 misfolding	R175H, R273H, R273C, R282W	I	2016 [53]
**SAHA** **(Vorinostat)**	HDAC inhibitor	283	Inhibition of the Hsp90 complex- induction of degradation of the mutant p53	R249S, R273H	I–III	2011 [54]
**PEITC**	Phytochemical	7	Restoration of the native conformation Oxidative stress	R175H	I–III	2016 [55]

**Table 2 ijms-23-13287-t002:** Currently registered clinical trials for the use of APR-246 in oncology patients.

ClinicalTrials.gov Identifier	Status	Conditions	Last Update	Available Results	Phase
**NCT04214860**	Completed	Myeloid Malignancy	19 January 2022	No	I
**NCT03931291**	Completed	Acute Myeloid Leukemia (AML), Myelodysplastic Syndromes (MDS)	19 January 2022	Yes [68]	II
**NCT04383938**	Completed	Bladder Cancer, Gastric Cancer, Non-Small Cell Lung Cancer (NSCLC), Urothelial Carcinoma, Advanced Solid Tumor	3 June 2022	Yes [69]	I–II
**NCT03588078**	Unknown	MDS with gene mutations, AML with gene mutations, Myeloproliferative Neoplasm (MPNs), Chronic Myelomonocytic Leukemia (CMML)	30 January 2020	Yes [72]	I–II
**NCT04419389**	Suspended	Non-Hodgkin Lymphoma (NHL), Chronic Lymphocytic Leukemia (CLL), Mantle Cell Lymphoma (MCL)	3 June 2022	No	I–II
**NCT03745716**	Completed	MDS with MUTp53	12 July 2022	Yes [https://clinicaltrials.gov/ct2/show/NCT03745716 accessed on 2 October 2022]	III
**NCT03268382**	Completed	High-grade Serous Ovarian Cancer (HGSC) with MUTp53	21 July 2022	No	II
**NCT03072043**	Completed	MDS, AML, MPNs; CMML with MUTp53	24 January 2022	Yes [73]	I–II
**NCT00900614**	Completed	Hematologic Neoplasms, Prostatic Neoplasms	31 July 2019	No	I
**NCT02098343**	Completed	HGSC with MUTp53	13 October 2022	Yes [71]	I–II
**NCT03391050**	Terminated	Melanoma	31 July 2019	No	I–II
**NCT04990778**	Withdrawn	MCL	10 March 2022	No	II

**Table 3 ijms-23-13287-t003:** Clinical trial for the use of COTI-2 in oncology patients.

ClinicalTrials.gov Identifier	Status	Conditions	Last Update	Available Results	Phase
**NCT02433626**	Unknown	Ovarian Cancer, Fallopian Tube Cancer, Endometrial Cancer, Cervical Cancer, Peritoneal Cancer, Head and Neck Cancer (HNSCC), Colorectal Cancer, Lung Cancer, Pancreatic Cancer	1 February 2019	No	I

**Table 4 ijms-23-13287-t004:** Clinical trial for the use of SAHA in oncology patients.

ClinicalTrials.gov Identifier	Status	Conditions	Last Update	Available Results	Phase
**NCT00735826**	Completed	Aerodigestive Tract Cancer, Lung Cancer, Esophageal Cancer, Head and Neck Cancer (HNSCC)	12 October 2018	No	NA
**NCT02538510**	Completed	HNSCC, Squamous Cell Carcinoma, Nasopharynx Carcinoma, Salivary Gland Carcinoma	13 September 2022	Yes [87]	I–II
**NCT00616967**	Active, Not Recruiting	Breast Cancer	3 February 2022	Yes [88]	II
**NCT01153672**	Completed	Breast Cancer	6 September 2019	Yes [https://clinicaltrials.gov/ct2/show/NCT01153672 accessed on 2 October 2022]	NA
**NCT00967057**	Completed	Leukemia	12 August 2013	Yes [89,90]	III
**NCT00121225**	Completed	Melanoma	29 January 2019	Yes [91]	II
**NCT00948688**	Terminated	Pancreatic Cancer	10 May 2017	No	I–II
**NCT01075113**	Completed	Liver Cancer	20 August 2019	Yes [92]	I
**NCT02042989**	Completed	Advanced Cancers with MUTp53	11 July 2022	No	I
**NCT01738646**	Completed	Glioblastoma	6 March 2017	Yes [93]	II

NA—not applicable.

**Table 5 ijms-23-13287-t005:** Clinical trial for the use of PEITC.

ClinicalTrials.gov Identifier	Status	Conditions	Last Update	Available Results	Phase
**NCT03700983**	Completed	Head and Neck Cancer	9 October 2018	No	NA
**NCT03034603**	Active, Not Recruiting	Head and Neck Neoplasms	10 October 2022	No	NA
**NCT00691132**	Completed	Lung Cancer	12 May 2017	Yes [100]	II
**NCT00968461**	Withdrawn	Leukemia	15 April 2013	No	I
**NCT01790204**	Completed	Oral Cancer with MUTp53	23 March 2015	No	I–II
**NCT00005883**	Completed	Lung Cancer	28 March 2011	No	I
**NCT02468882**	Unknown	Long-term Effects Secondary to Cancer	11 June 2015	No	III
**NCT03978117**	Recruiting	Healthy	5 April 2022	No	II
**NCT05354453**	Recruiting	Healthy	19 October 2022	No	I
**NCT05070585**	Recruiting	Metabolic Disturbance	9 June 2022	No	I–II

NA—not applicable.

## Data Availability

Not applicable.

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
