# Peer review of "Gain of Function (GOF) Mutant p53 in Cancer—Current Therapeutic Approaches"

_ijms, 2022, doi:10.3390/ijms232113287_

Round 1

Reviewer 1 Report

This review aims at describing the state of the art in the clinical application of drug designed to target missense mutant p53 in cancer. The idea is enticing, but this manuscript is inadequate.

There are a number of recent reviews describing pharmacological approaches to target mutp53 – some of them quite detailed, and some also critical regarding the assumption that targeting mutant p53 would really benefit cancer therapy (e.g. Duffy et al., Cancers 2022; Levine, Cancer Res 2022; Bykov et al., Nat Rev Cancer 2018; Sabapathy and Lane. Nature Reviews Clinical Oncology 2018; Wang et al., Cell Death Differ 2022).

The present manuscript does not provide any new information, nor does it offer an innovative perspective towards the interpretation of current evidences.

The information provided on clinical trials is pointless: authors only list many trial numbers, as downloaded from the www.clinicaltrials.gov database. The only mention of possible results of a clinical trial is at line 320 (on the combination of eprenetapopt and azacytidine).

Moreover, the introductory part denotes incomplete comprehension of the many facets of the mutant p53 gain-of-function and its regulation. Finally, the text is often confusing. 

A few examples

Line 43. “Another regulation is possible due to its lack of chaperones or activators.” 

The meaning of this sentence is unclear. 

Line 192 -197. This paragraph is not informative. It is just a list of cell lines with different mutations of various genes; there is no info on the type of p53 mutation and - importantly – there is no information or comment on the results (!).

Line 236. “Several studies suggest direct reaction between SAHA and p53 in cases where apoptosis was a result of p53 activation via phosphorylation”

This sentence is misleading – it implies that SAHA acts on p53. To my knowledge there is no evidence in such direction

Line 315. “when partial loss in activity of p53 is introduced, cancer risk is increased. However, when there is minimal to no activity at all, the phenomena known as p53 addiction [34,83,84] occurs, which in turn causes the tumor to be reduced just by small activity that appeared due to the applied drug [85].”

This sentence is quite confusing – especially being part of the concluding remarks

Author Response

Please see the attachament

Reviewer 2 Report

I regret to say that the manuscript in the present form is not suitable for publication.

The 2 main reasons are:

A very poor use of English and several typos and nomenclature inconsistencies that makes the reading very hard.

Some examples are:

Line 24: “It is implied that 50% of oncological patients acquire…”. Not sure if “implied” is the right word and the meaning is hard to understand

Line 27: “TP53 mutations demonstrate themselves either as Loss-of-Funtion…”. “Demonstrate” should be changed and loss-of-function should not be capital.

Line 28: “GOF TP53 mutants convey increased malignancy…”. Convey should be changed to “confer”.

The mutant p53 is variably indicated as MUTp53 (line 78), mutp53 (line 145), mut p53 (line 106).

In line 147 it is not clear if MQ-C124 and MQC229 refers to mutations in p53 and the notation for mutations is different from the one used in the beginning of the sentence “277-cysteine”.

In many other sentences the subject is not clear, the article is absent, the punctuation is not properly used.

Many, if not all the paragraphs appear as a collection of findings which are not adequately put into context and for which it is difficult to find a logic thread that connects them.

I would suggest the following:

to reorganize the manuscript and the different sections to improve the internal lack of cohesion

to provide some figures that could help the reader navigating the factual evidence that is mentioned in the manuscript. For example in relation to the different physiological roles of p53 illustrated in lines 34 – 55 and the different types of chemoresistance of different p53 mutant mentioned in lines 98 – 125.

A thorough language check that could bring the English level to an acceptable standard for an international publication.

Reviewer 3 Report

This review intended to focus on gain of function mutant p53 in cancer and discuss pharmacological therapeutic approaches targeting mutant p53 exemplified by four molecules that in clinical trials. There are many similar reviews published on the same topic. This review needs to be improved to provide either a novel perspective or more recent development in the field. Thus, the manuscript may not be suitable for publication in IJMS unless the following concerns are addressed.

Section 1. Introduction

Given that the focus of this review is on p53 mutations, it would be useful to describe the structure of p53, for instance by using a schematic representation of its main domains. This might aid comprehension of the GOF p53 mutants described in the subsequent sections.

Section 2.

Besides Puma, Bax and Noxa, Death receptor 5 (DR5) is also an important target gene of p53, which regulates cellular apoptosis in response to cellular stresses.

In addition to MDM2, MDM4 (MDMX) also negatively regulates p53 transcriptional activity by interacting with MDM2. This regulatory mechanism maintains basal p53 levels under physiological conditions. This should be included in section 2.

Section 3.

If the authors introduce more information on p53 structures, the common GOF mutations in p53 can be described more clearly for example within the DNA binding domain.  

Section 5.

As the title mentioned, may the authors elaborate on the current strategies targeting p53 including elimination of mutant p53, restoration or reactivation of wtp53 and others? This could help readers better understand the mechanisms of action of the selected molecules described in followed the sections.

What is the criterion or rationale for selecting the four compounds (APR-246, COTI-2, SAHA, PEITC) for this review? If the authors intended to consider drugs in clinical trials, the list is not complete. For example, ALRN-6924, a dual MDM4 and MDM2 inhibitor, is also in multiple clinical trials as monotherapy or as part of a combination in different tumors (NCT02909972, NCT02264613, etc.). Concurrent inhibition of MDM4 and MDM2 is preferable than inhibition of MDM2 alone, especially in the tumors with MDM4 overexpression. Other examples of molecules that target p53 and enter clinical trials include milademetan, RG7112, RG7388, CGM097, HDM201 and many others. The authors need to either convince the readers the importance of these four molecules or include other molecules in the discussion.

Sections 6-9.

COTI-2 is effective against all types of p53 mutation, including R175H, R273H, R273C, R282W (Breast Cancer Res Treat, 2020, 179, 47; Clin Cancer Res 2019, 25, 5650). However, R175H is the only targeted p53 mutant of COTI-2 listed in Table 1. The authors should keep the literature up to date.  

It may be more readable to tabulate the clinical trial codes for each molecule, other than describing them in the sections.

Round 2

Reviewer 1 Report

Authors made a superficial attempt to improve their original manuscript. They fixed some of the problems, and listed the clinical trials in tables, which is a significant improvement. Nonetheless, the information content and the overall contribution to the field remains very limited.

The manuscript is still not pleasant to read, the information is delivered in a chaotic way, and the text has several badly written sentences even in the revised version. I recommend that the authors ask someone else to re-read the review, and perform a careful proofreading of the text; not for spelling (which is fine), but for construction of the sentences. 

This is not the job of a reviewer, so I will point out just a few examples.

Line 117 - It was shown that WTp53 as well as p53 carries GOF missense mutations can inhibit the expression of SLC7A11 and sensitize cells to ferroptosis. 

Line 135 - Chemoresistance in GOF MUTp53 manifests itself differently in each type of p53 mutation due to different proteins and genes being affected as a cause of a completely distinct mechanism.

Line 279 - Successful in a variety of cancers in in vitro research (lymphoma, etc...) and well tolerated by the patients diagnosed with cutaneous T-cell lymphoma establishes its multipurpose use.

One last point. In the legend to Figure 1, the drawing is a scheme of the domain structure of p53. Not its secondary structure. The secondary structure of a protein is something completely different. 

Reviewer 2 Report

The manuscript has been significantly improved and it is now suitable for publication

Reviewer 3 Report

The revised manuscript has been significantly improved. I have no further questions.
